# Cellular Environment Remodels the Genomic Fabrics of Functional Pathways in Astrocytes

**DOI:** 10.3390/genes11050520

**Published:** 2020-05-07

**Authors:** Dumitru A Iacobas, Sanda Iacobas, Randy F Stout, David C Spray

**Affiliations:** 1Personalized Genomics Laboratory, Center for Computational Systems Biology, RG Perry College of Engineering, Prairie View A&M University, Prairie View, TX 77446, USA; 2DP Purpura Department of Neuroscience, Albert Einstein College of Medicine, New York, NY 10461, USA; 3Department of Pathology, New York Medical College, Valhalla, NY 10595, USA; sandaiacobas@gmail.com; 4Department of Biomedical Sciences, College of Osteopathic Medicine, New York Institute of Technology, Old Westbury, NY 11568, USA; rstout@nyit.edu; 5Department of Medicine, Albert Einstein College of Medicine, New York, NY 10461, USA; david.spray@einsteinmed.org

**Keywords:** calcium signaling, chemokine signaling, Cx43, gap junction, NOD-like receptor signaling, oli-neu cells, pannexin1, PI3K-Akt pathway, thyroid hormone pathway

## Abstract

We profiled the transcriptomes of primary mouse cortical astrocytes cultured alone or co-cultured with immortalized precursor oligodendrocytes (*Oli-neu* cells). Filters between the cell types prevented formation of hetero-cellular gap junction channels but allowed for free exchange of the two culture media. We previously reported that major functional pathways in the *Oli-neu* cells are remodeled by the proximity of non-touching astrocytes and that astrocytes and oligodendrocytes form a panglial transcriptomic syncytium in the brain. Here, we present evidence that the astrocyte transcriptome likewise changes significantly in the proximity of non-touching *Oli-neu* cells. Our results indicate that the cellular environment strongly modulates the transcriptome of each cell type and that integration in a heterocellular tissue changes not only the expression profile but also the expression control and networking of the genes in each cell phenotype. The significant decrease of the overall transcription control suggests that in the co-culture astrocytes are closer to their normal conditions from the brain. The *Oli-neu* secretome regulates astrocyte genes known to modulate neuronal synaptic transmission and remodels calcium, chemokine, NOD-like receptor, PI3K-Akt, and thyroid hormone signaling, as well as actin-cytoskeleton, autophagy, cell cycle, and circadian rhythm pathways. Moreover, the co-culture significantly changes the gene hierarchy in the astrocytes.

## 1. Introduction

The main glial cells, astrocytes, and oligodendrocytes, together with ependymal cells and microglia, form what has been called the “silent majority” of the cells of the brain. These cells were considered “silent” because they do not generate action potential as neurons do. Nonetheless, they do signal among themselves by exchanging small (<1 kD) molecules via gap junction channels and release of gliotransmitters that bind the membrane receptors of the cells in the neighborhood. Astrocyte and oligodendrocyte interactions, among themselves and with each other, form the panglial network linking these glial cells throughout the brain. The panglial network provides metabolic support for neuronal activity, thereby impacting both constitutive brain functions such as sleep but also dynamic activities that include learning and cognition [1,2,3].

Gap junctions provide a bidirectional route for interaction between oligodendrocytes and astrocytes [4,5,6], where specific connexin proteins form the intercellular channels in the different cells. Astrocytes express Cx43 (*Gja1*) and Cx30 (*Gjb6*), whereas oligodendrocytes express Cx32 (*Gjb1*) and Cx47 (*Gja12*), and the two cell types communicate through heterotypic connexin pairing (Cx43:Cx47, Cx30:Cx32) [7]. As we have shown in a 2003 paper [8], knocking out Cx43 alters numerous genes in the mouse cortical astrocytes. None of the connexins expressed in astrocytes or oligodendrocytes is found in brain neurons (that express Cx36 and, to a lesser extent, Cx45). We previously examined gene expression patterns in brains of mice lacking the genes encoding the astrocyte gap junction protein Cx43, the oligodendrocyte gap junction protein Cx32 or the neuron gap junction protein Cx36 (*Gjd2*) compared to wildtype counterpart mice [9,10,11,12,13]. These studies revealed that the brain transcriptomes of Cx43KO mouse and Cx32KO mouse were both 83% altered with respect to the wildtype counterpart, whereas their similarity to the brain of Cx36KO mouse were respectively 12% and 11% [12]. Since astrocytes and oligodendrocytes form heterocellular gap junction channels with each other [14,15,16], but none with neurons has been yet reported in the mouse brain (after our knowledge), these results suggested “panglial transcriptomic continuity”. By transcriptomic continuity we understand that transcriptomic changes in one cell type has consequences in another cell type. Panglial transcriptomic continuity is consistent with the disrupted expression of *Gjb1*, the oligodendrocyte coupling partner of the astrocyte *Gjb6* (encoding Cx30) in brains of Cx30 knock-out (KO) mice [17]. It can also explain the reduction of astrocyte Cx43 in the EAE model of multiple sclerosis that targets production of myelin protein by oligodendrocytes [18].

Astrocytes and oligodendrocytes actively modulate the chemical environment of all brain cells. Thus, astrocytes release ATP [19,20,21] (to mediate Ca^2+^-signaling among glial cells [22] and in response to sleep-pressure [23]), glutamate to control synaptic strength [24] and several cytokines and chemokines [25,26,27]. Glial dysfunction is responsible for a wide spectrum of neurological diseases (e.g., [28,29,30,31]).

In previously published studies [32,33], we determined gene expression changes induced in an immortalized oligodendrocyte precursor cell line (*Oli-neu* cells [34]) co-cultured with but not in contact with astrocytes so as to assure that effects were not mediated by adhesive or gap junctions. That study revealed substantial impact of astrocyte proximity on several functional pathways in *oli-neu* cells, with major up-regulation in myelination and its regulation by calcium signaling and cytokine interactions with their receptors [32,33]. *Oli-neu* cells are representative of immature oligodendrocytes. However, in terms of expression of myelin proteolipid protein (*Plp1*), myelin basic protein (*Mbp*), and 2’,3’-cyclic nucleotide 3’-phosphodiesterase (*Cnp*), *Oli-neu* cells appear to be much more differentiated than other immortalized precursor oligodendrocytes (like *N20.1* cells [35]).

In the present study, we tested the extent to which the astrocyte-oligodendrocyte interactions are bidirectional by comparing the transcriptomes of mouse cortical astrocytes cultured alone or co-cultured with *Oli-neu* cells in insert systems that prevented formation of hetero-cellular gap junction channels [36] but allowed free exchange of the two culture media. Results show that the proximity of *Oli-neu cells* induced changes in astrocyte transcriptome involving major functional pathways, including those underlying calcium, PI3K-Akt, chemokine, thyroid hormone, and NOD-like receptor signaling. 

## 2. Materials and Methods 

### 2.1. Cells 

Primary cortical astrocytes were isolated as previously described [37] from meninges-free brains of twelve mouse pups obtained though caesarian section of day 19 pregnant C57Bl/6j mice. Astrocytes from each mouse were collected in separate vials and about 91–92% of the cells were immuno-positive for glial acidic fibrillary protein (GFAP Neurons, oligodendrocytes and microglial cells were absent. Animals were housed in the Animal Facility of the Albert Einstein College of Medicine and procedures were performed according to the IACUC approved procedure (current renewal 20180816, approved 2018). Brains, minced separately in 500 μL of 0.05% trypsin-EDTA, were transferred to 1 mL Eppendorf tubes containing 500 μL Dulbecco’s Modified Eagle medium (DMEM) supplemented with 10% fetal bovine serum, 1% penicillin-streptomycin and spun down by centrifugation at 450 g for 10 min. Cells from each brain were resuspended in culture medium, plated in 100 mm culture dishes and maintained in a humidified 5% CO_2_ incubator at 37 °C. After 1 week, the primary confluent astrocyte culture from each mouse was separately trypsinized and re-plated. Astrocytes from each mouse were maintained at confluence during the entire experiment. 

The *Oli-neu* cell line was generated by retroviral transduction of mouse oligodendrocyte precursors with the *t*-neu oncogene [34,38] and kindly provided by Dr. J. Trotter (University of Mainz, Germany). The cells were seeded and grown on poly-L-lysine-coated culture dishes in DMEM medium + 1% B27, N2, PSG, SP, and 1% horse serum (GIBCO^®^, Waltham, MA, USA) maintained at 37 °C in 5% CO_2_ in a humidified incubator, and passaged following dissociation with Trypsin-EDTA.

### 2.2. Experimental Arrangement 

The cells were plated as previously described [33] in FalconTM cell culture 6 well insert systems (www.fishersci.ca), with cortical astrocytes from separate mice in all plates. *Oli-neu* cells were placed in all the six inserts of the first system and only the culture medium used for *Oli-neu* cells (but without any cells present) in all the six inserts of the second system. Owing to astrocytes adhering to the bottom of the companion plate and *Oli-neu* cells confined to the insert, formation of hetero-cellular gap junction channels between astrocytes and *Oli-neu* cells [32] was prevented. However, the astrocytes were exposed to the molecules released by the oligodendrocytes diffusing through the 0.4 µm pores of the inserts. The astrocytes were collected in separate labeled Eppendorf vials after 10 days in the Falcon systems. Four vials with well-developed astrocytes were selected from each of the two Falcon systems. Since in each experiment the astrocytes came from a different mouse, we had four independent biological replicas (labeled **INS**) with astrocytes co-cultured with *Oli-neu* cells and four biological replicas (labeled **CTR**) with astrocytes in *Oli-neu* culture medium alone. 

### 2.3. Microarray 

Total RNA was extracted as previously described [39] with Qiagen RNeasy minikits separately from each of the eight selected vials from the two Falcon systems. RNA concentration before and after reverse transcription in the presence of Cy3/Cy5 dUTP was determined with NanoDrop ND 2000 Spectrophotometer and quality with Agilent, (Santa Clara, CA, USA) RNA 6000 Nano kit in an Agilent 2100 Bioanalyzer. 825 ng of differently (Cy3/Cy5) labeled biological replicas were hybridized 17 h at 65 °C with Agilent G2519F unrestricted AMADID Release GE 4 × 44k 60 m two-color mouse gene expression microarrays using the “multiple yellow” strategy. The chip (4 microarrays) was scanned with an Agilent G2539A dual laser scanner at 5 μm resolution in 20-bit scan mode (>10^5^ dynamic range) and primary analysis performed with (Agilent) Feature Extraction 11.6 software. 

All corrupted spots or with foreground fluorescence less than twice background fluorescence in any of the eight samples were eliminated from the analysis. Data were normalized using our standard algorithm alternating intra- and inter-array normalization to the median of the background-subtracted fluorescence. Spots probing the same transcript were grouped into redundancy groups. Agilent mouse 4 × 44k microarrays used in this study hybridizes 30,175 distinct transcripts, out of which 22,657 are probed by single spots. The largest redundancy groups (13 spots) probed the genes: *Abcc5* (ATP-binding cassette, sub-family C (CFTR/MRP), member 5)*, Cpne4* (copine IV)*, Csf1* (estrogen receptor 1 colony-stimulating factor 1)*, Esr1* (estrogen receptor 1)*, Mapk1* (mitogen-activated protein kinase 1)*, Oprm1* (opioid receptor, mu 1)*, P2rx3* (purinergic receptor P2X, ligand-gated ion channel, 3), and *Socs2* (suppressor of cytokine signaling 2).

### 2.4. Data Analysis 

Profiling four biological replicas of each condition produces with adequate statistical power three independent measures for each transcript: (i) average expression level, (ii) variability of transcript abundance, and (iii) expression coordination with each other transcript [40]. The rarely used analysis of expression variability provides information about the degree to which homeostatic mechanisms limit range of transcript abundance and the analysis of expression coordination allows assessment of interactions within gene networks that underlie functional pathways. We report here the astrocyte genes that were up-or down-regulated, exhibited stricter or looser expression control, and were differently networked when the *Oli-neu* cells are close by. The medians of the three independent gene features can be used to characterize selected groups of genes and transcriptomic networks associated with functional pathways.

### 2.5. Pathway Analysis 

Kyoto Encyclopedia for Genes and Genomes [41,42] was used to select the genes responsible for calcium (map mmu04020), PI3K-Akt (mmu04151), chemokine (mmu040602), thyroid hormone (mmu04919) and NOD-like receptor (mmu04621) signaling pathways. We have also studied the remodeling of the actin cytoskeleton (mmu04810), autophagy (mmu04140), cell-cycle (mmu04110), circadian rhythm (mmu4710), and gap junction (mmu04540). Particular attention was given to the regulation of the astrocytic receptors involved in modulating the activity of the glutamatergic (mmu04724), GABAergic (mmu04727), cholinergic (mmu04725), dopaminergic (mmu04728) and serotonergic (mmu04726) interneuron synapses. 

These pathways were selected for the following reasons:-Calcium signaling (hereafter denoted by **CAS**) is evolutionary the oldest, yet most common way by which a wide diversity of cells communicates to each other [22]. Change in calcium signaling is a major modulator of the glial cell behavior [43].-PI3K-Akt signaling (hereafter denoted by **PA**) is pivotal for the growth, metabolism, survival, angiogenesis, autophagy, and chemotherapy resistance of the malignant astrocytic glioma [44].-Chemokine signaling (**CS**) between astrocytes and oligodendrocytes, most likely the main crosstalk in our experiment, is important for glial development and stimulating regeneration and repair [45]. -The thyroid hormone signaling pathway (**TH**) was chosen because astrocytes are thought to be the main regulator of thyroid hormone in the brain and T3 is a main driver of oligodendrocyte maturation [46,47].-NOD-like receptor signaling pathway (**NOD**) was chosen because of its role in cognition, anxiety, and activation of the hypothalamic-pituitary-adrenal axis [48]. -The actin cytoskeleton (**AC**), is an elaborate cytoplasmic protein structure central in determining cell and organ size and morphology, intracellular transport and cell division [49].-Autophagy (**AU**) is a major degradation pathway, essential in maintaining astrocyte function [50].-Cell-cycle (**CC**) is expected to be one of the most dependent pathways in the cellular environment. -The circadian rhythm (**CR**)–increasing evidence indicates that astrocytes are very important players in the regulation of circadian rhythms [51].-Even though the experimental set up did not allow formation of hetero-cellular gap junction channels, the soluble factors secreted by the other cell type might have an effect on the expression level and networking of the gap junction (**GJ**) pathway in astrocytes.

### 2.6. Relative Expression Variability 

Expression variability is normally quantified by the coefficient of variation (CV). However, owing to the non-uniform redundancy of the microarray spots probing the same transcript), we use a Bonferroni-like corrected p-val < 0.05 significant chi-square mid-interval estimate of CV for each distinct transcript in each condition which we term Relative Expression Variability (REV) [40]:
(1)$$\begin{array}{l} REV_{i}^{\left(condition\right)}=\underset{\mathrm{correction}\;\mathrm{coefficient}}{\underbrace{\frac{1}{2}\left(\sqrt{\frac{r_{i}}{\chi^{2}\left(r_{i};0.975\right)}}+\sqrt{\frac{r_{i}}{\chi^{2}\left(r_{i};0.025\right)}}\right)}}\sqrt{\underset{\mathrm{pooled}\;\mathrm{CV}}{\underbrace{\frac{1}{R_{i}}\sum_{k=1}^{R_{i}}{\left(\frac{s_{ik}^{\left(condition\right)}}{\mu_{ik}^{\left(condition\right)}}\right)}^{2}}}}\times 100\%\\ \mu_{ik}=\mathrm{average}\;\mathrm{expression}\;\mathrm{level}\;\mathrm{of}\;\mathrm{gene}\;i\;\mathrm{probed}\;\mathrm{by}\;\mathrm{spot}\;k\;(=1,\;\ldots ,\;R_{i})\;\mathrm{in}\;\mathrm{the}\;4\;\mathrm{biological}\;\mathrm{replicas}\\ s_{ik}=\mathrm{standard}\;\mathrm{deviation}\;\mathrm{of}\;\mathrm{the}\;\mathrm{expression}\;\mathrm{level}\;\mathrm{of}\;\mathrm{gene}\;i\;\mathrm{probed}\;\mathrm{by}\;\mathrm{spot}\;k\\ r_{i}=4R_{i}-1=\;\mathrm{number}\;\mathrm{of}\;\mathrm{degrees}\;\mathrm{of}\;\mathrm{freedom}\\ R_{i}=\mathrm{number}\;\mathrm{of}\;\mathrm{microarray}\;\mathrm{spots}\;\mathrm{probing}\;\mathrm{redundantly}\;\mathrm{gene}\;i \end{array}$$

### 2.7. Expression Regulation

A gene was considered to be significantly regulated if its absolute fold-change |*x*| exceeded the corresponding individual gene cut-off (*CUT*) that considers the combined contributions of the biological variability among biological replicas and the technical noise in achieving the measurement. This method eliminates most of the false positive and negative hits that result from use of an arbitrary fixed cut-off (such as 1.5x) [52].
(2)$$\begin{array}{l} \left|x_{i}\right|>CUT_{i}=1+\frac{1}{100}\sqrt{2\left({\left(REV_{i}^{\left(INS\right)}\right)}^{2}+{\left(REV_{i}^{\left(CTR\right)}\right)}^{2}\right)}\;,\;where:\\ x_{i}=\{\begin{array}{cc} \frac{\mu_{i}^{\left(INS\right)}}{\mu_{i}^{\left(CTR\right)}} & ,\;\mathrm{if}\;\mu_{i}^{\left(INS\right)}>\mu_{i}^{\left(CTR\right)}\\ -\frac{\mu_{i}^{\left(CTR\right)}}{\mu_{i}^{\left(INS\right)}} & ,\;\mathrm{if}\;\mu_{i}^{\left(INS\right)}<\mu_{i}^{\left(CTR\right)} \end{array} \end{array}$$

### 2.8. Pathway Regulation 

The regulation of a given pathway was analyzed from the perspective of both percent of genes that were significantly regulated (using the above criterion of the absolute individual gene fold-change cut-off) and the Weighted Pathway Regulation [40]:
(3)$$WPR(\Gamma )={\langle \mu_{i}^{\left(CTR\right)}\times \left(\left|x_{i}\right|-1\right)\times \left(1-p_{i}\right)\rangle }_{i\in \Gamma }\times 100$$

In (3), *p_i_* is the *p*-value of the heteroscedastic (two tails, unequal variance) *t*-test of the means equality in the two conditions.

WPR quantifies the contributions of the composing genes to the pathway alteration by considering their normal expression levels (here in CTR), their absolute departure from equal expression in both conditions, and the statistical significance of their regulation. 

### 2.9. Expression Correlation 

Pair-wise Pearson product-moment correlation analysis of the (log_2_) expression levels across the biological replicas was performed to identify the significantly (p-val < 0.05) synergistically, antagonistically, and independently expressed gene pairs in each condition. Two genes were considered as (*p* < 0.05) significantly synergistically expressed (*ρ* > 0.95) when their expression levels are positively correlated across biological replicates, while they are antagonistically expressed (*ρ* < −0.95) when their expression levels manifest opposite tendencies. The genes are considered as independently expressed when (|*ρ*| < 0.05). The statistical significance of the correlation coefficient was determined with the two-tail *t*-test for the degrees of freedom df = 4(biological replicas)*R (number of spots probing redundantly each of the correlated transcripts) – 2. [53]. The correlation analysis was used to determine the remodeling of gene networks.

### 2.10. Gene Commanding Height 

The Gene Commanding Height (GCH) score was introduced recently [54,55] to establish the gene hierarchy in each condition. It combines an estimate of the transcription control of that gene with a measure of its expression coordination with each other gene:
(4)$$\begin{array}{l} GCH_{i}^{\left(condition\right)}=\underset{\mathrm{transcription}\;\mathrm{control}\;\mathrm{estimate}}{\underbrace{\frac{\langle REV^{\left(condition\right)}\rangle }{REV_{i}^{\left(condition\right)}}}}\times \underset{\mathrm{measure}\;\mathrm{of}\;\mathrm{expression}\;\mathrm{coordination}}{\underbrace{\exp\left(4{\bar{{\left(\rho_{ij}^{\left(condition\right)}\right)}^{2}}|}_{\forall j\neq i}\right)}},\;where:\\ \langle REV^{\left(condition\right)}\rangle =\mathrm{median}\;\mathrm{of}\;\mathrm{all}\;\mathrm{genes}\;\mathrm{REVs}\;\mathrm{in}\;“\mathrm{condition}”\;\mathrm{CTR}\;\mathrm{or}\;\mathrm{INS}\\ \rho_{ij}^{\left(condition\right)}={\mathrm{Pearson}\;\mathrm{correlation}\;\mathrm{coefficient}\;\mathrm{of}\;\mathrm{genes}\;i\;\mathrm{and}\;j\;\log}_{2}\;\mathrm{expression}\;\mathrm{levels}\\ \; \end{array}$$

The top gene (highest GCH) was termed the Gene Master Regulator (GMR). The strong transcription control indicates that the right abundance of the GMR transcript is critical for the cell phenotypic expression, while the high expression coordination indicates that the power of the GMR to regulate the functional pathways. 

## 3. Results

### 3.1. Changes in Morphology of Oli-New Cells 

We observed that, Oli-neu cells became more differentiated after 10 days in co-culture compared to culture in the absence of astrocytes (illustrated in Appendix A). Differentiation of the Oli-neu cells, similar to progression from the stage of precursor oligodendrocytes to preoligodendrocytes, was visually evaluated by comparing the MAG (myelin-associated glycoprotein)–labeled phase-contrast microphotographs of the cells cultured alone and in co-culture. Our previously reported transcriptomic analyses [32,33] also indicate differentiation of the Oli-neu cells in culture, reflected in the activation of the myelination-related components of the genomic fabric and its control by the cytokine receptors and calcium signaling pathway. 

### 3.2. Overview of the Microarray Data

Raw and processed gene expression data were deposited in the publicly available website https://www.ncbi.nlm.nih.gov/geo/query/acc.cgi?acc=GSE109035. In total, 18,891 unigenes were adequately quantified in all 8 samples, out of which almost 20% were significantly regulated in astrocytes cultured in the proximity of *Oli-neu* cells (INS) compared to astrocytes cultured alone (CTR). 

The analyzed pathways were composed of the following numbers of unigenes: **AC** = actin-cytoskeleton (181 unigenes), **AU** = autophagy (122), **CAS** = calcium signaling (137), **CC** = cell cycle (116), **CR** = circadian rhythm (29), **CS** = chemokine signaling (139), **GJ** = gap junction (78), **NOD** = NOD-like receptor signaling (143), **PA** = PI3K-Akt signaling (294), **TH** = thyroid hormone signaling (106). In addition to these pathways, we have considered also a group of 22 astrocyte genes, termed **sy**napse **r**egulators (**SYR**), that are considered by the KEGG analysis to modulate the synaptic transmission between neurons in the brain.

An overview of the microarray results is presented in Figure 1. Thus, Figure 1a shows the alterations of the expression level, Figure 1b the alterations of Relative Expression Variability (REV) and Figure 1c the changes in the Gene Commanding Height (GCH) for all quantified unigenes (ALL) and each of the selected functional pathways. In addition, Figure 1d presents the GCH scores of the top 5 genes in each condition and the corresponding GCH in the other condition. 

Alterations of the expression level are presented as percentages of the up- and down-regulated genes, and as the Weighted Pathway Regulation (WPR). The individual gene cut-off (*CUT*) ranged from 1.06× for TGF-beta activated kinase 1/MAP3K7-binding protein 1 (*Tab1*) from NOD, up-regulated by 1.25×, to 3.61× for guanylate cyclase 1, soluble, alpha 2 (*Gucy1a2*) from GJ, significantly up-regulated by 35.94×. Using the uniform 1.5× absolute fold-change would result in 4% false hits and neglect 6% significant regulations as a consequence of sample variability.

Appendix A lists the genes whose >1.5× absolute fold-change did not meet the individual CUT criterion (“false hits” for the uniform 1.5× fold-change cut-off). Appendix A list the genes considered as significantly up- and down-regulated according to our criterion, although some of them were below the 1.5× traditionally used as a cut-off.

For the entire transcriptome, the percentage of down-regulated genes is balanced by that of the up-regulated (9.63% down- vs. 9.89% up-regulated, down/up ratio = 0.97). However, the percentages of the down- to up-regulated genes are quite dissimilar for individual pathways, the most notable example being the cell cycle (CC) pathway where the ratio of down to up-regulated genes is 31×. The bias indicates that proximity of the *Oli-neu* cells had a profound slow-down effect on this pathway in astrocytes (detailed regulation in Section 3.3.2). 

Note in Figure 1b that REV was higher in co-cultured astrocytes for all gene groups except those related to the chemokine signaling (CS) and gap junction (GJ) pathways and was statistically significant for ALL, AC, AU, CAS, PA, and TH groups. Figure 1c shows that mean GCH was significantly higher for the synaptic receptors, gap junction, autophagy, and chemokine signaling, indicating increased role of these pathways in the co-cultured astrocytes. GCH values were significantly lower for the circadian rhythm pathway, while the differences of the other pathways were not statistically significant. It is interesting to observe in Figure 1d that not only is there no overlap of the top 5 genes in the two conditions, but that a highly ranked gene in one condition has a very low score in the other. This difference in GCH indicates that the effect of the cellular environment goes far beyond the expression regulation of some genes, affecting also the control of the transcript abundance and gene networking. Thus, a gene that plays a major role in one condition may play a minor one in the other. 

The gene with the highest GCH in the CTR astrocytes was *Serp2* (stress-associated endoplasmic reticulum protein family member 2) with GCH = 49.31 (GCH = 2.49 in INS astrocytes), while in the INS astrocytes it was *Pdcd7* (programmed cell death 7), with GCH = 48.13 (compared to GCH = 2.14 in CTR astrocytes). 

### 3.3. Regulation of Gap Junction, Cell-Cycle, Actin-Cytoskeleton, and Circadian Rhythm Pathways

Figure 2 and Figure 3 show the regulation of the KEGG-determined gap junction and cell-cycle pathways, and Appendix A the regulation of the actin-skeleton and circadian rhythm pathways. Graphs below pathways present the expression ratios (together with the respective individual gene fold-change cut-offs, both negative for down-regulation) of significantly altered genes. Each figure also shows the GCH scores of the pathway genes in the two conditions.

#### 3.3.1. Significantly Regulated Gap Junction-Associated Genes

Significantly up-regulated genes in the GJ pathway were gap junction protein, beta 2 (*Gjb2*, encoding Cx26), guanylate cyclase 1, soluble, alpha 2/beta 2 (*Gucy1a2/b2*), mitogen-activated protein kinase 3 (*Mapk3*), pannexin 1 (*Panx1*), tubulin, and beta 3 class III (*Tubb3*). 

Significantly down-regulated genes: adenylate cyclase 2 (*Adcy2*), cyclin-dependent kinase 1 (*Cdk1*), gap junction protein, gamma 1 (*Gjc1*, encoding Cx45), guanine nucleotide-binding protein, alpha q polypeptide (*Gnaq*), inositol 1,4,5-trisphosphate receptor 1 (*Itpr1*), platelet-derived growth factors (*Pdgfb, Pdgfc, Pdgfrb*), tubulin, beta 6 class V (*Tubb6*). Note that our method identified as significantly regulated *Adcy2*, *Gjc1*, *Gnaq*, *Gucy1b2*, *Itpr1*, *Mapk3*, *Panx1,* and *Pdgfc* that would be neglected by the fixed uniform 1.5× cut-off. Even though the individual CUT for *Gucy1a2* exceeded 1.5× due to high biological variability, the fold change was so large (32×) that the difference was significant. 

#### 3.3.2. Significantly Regulated Cell-Cycle Genes

As illustrated in Figure 3, the regulated (mostly down-regulated) genes are located in both DNA replication (S phase) and mitosis (M phase), separated temporally by the gaps G1 and G2. *Anapc2* (anaphase-promoting complex subunit 2), *Mad2l2* (mitotic arrest deficient-like 2) and *Tgfb1* (transforming growth factor, beta 1) were the only significantly up-regulated genes (however for all three the fold-change was less than 1.5×). 

Significantly down-regulated genes include cell division cycles (*Cdc20, Cdc25A, Cdc6*), cyclins (*Ccna2, Ccnb2, Ccnd1*), cyclin-dependent kinases (*Cdk2, Cdk6*), cyclin-dependent kinase inhibitor 2A (*Cdkn2a*), DBF4 homolog (*Dbf4*), E2F transcription factors (*E2f1, E2f2*), growth arrest and DNA-damage-inducible 45 (*Gadd45a, Gadd45b*), mini-chromosome maintenance deficient (*Mcm2*, *Mcm4, Mcm6, Mcm7*), transformed mouse 3T3 cell double minute 2 (*Mdm2*), origin recognition complex subunits (*Orc2, Orc6*), protein kinase, membrane-associated tyrosine/threonine 1 (*Pkmyt1*), polo-like kinase 1 (*Plk1*), RAD21 homolog (*Rad21*), retinoblastoma-like 1 (*Rbl1*), SMAD family member 2 (*Smad2*), structural maintenance of chromosomes 1A (Smc1a), transforming growth factor, beta 3 (*Tgfb3*), Ttk protein kinase (*Ttk*), WEE 1 homolog 1 (*Wee1*) and tyrosine 3-monooxygenase/tryptophan 5-monooxygenase activation protein, and eta polypeptide (*Ywhah*).

#### 3.3.3. Significantly Regulated Actin Cytoskeleton Genes

In the actin cytoskeleton pathway (Appendix A) the up-regulated genes were: chemokine (C-X-C motif) ligand 12 (*Cxcl12*), fibroblast growth factors (*Fgf1, Fgf9*), gelsolin (*Gsn*), integrins (*Itga9, Itgb2, Itgb4*), LIM-domain containing, protein kinase (*Limk1*), phosphatidylinositol-5-phosphate 4-kinase, type II, gamma (*Pip4k2c*), RAS-related C3 botulinum substrate 3 (*Rac3*), slingshot homolog 3 (*Ssh3*), WAS protein family, member 1 (*Wasf1*). 

Down-regulated genes were: actinin alpha 4 (*Actn4*), Rac/Cdc42 guanine nucleotide exchange factor 6 (*Arhgef6*), breast cancer anti-estrogen resistance 1 (*Bcar1*), Braf transforming gene (*Braf*), coagulation factor II (thrombin) receptor (*F2r*), guanine nucleotide-binding protein (G protein), gamma 12 (*Gng12*), insulin II (*Ins2*), IQ motif containing GTPase activating proteins (*Iqgap1, Iqgap2, Iqgap3*), integrins (*Itga3 Itgav Itgb1, Itgb5*), lysophosphatidic acid receptor 4 (*Lpar4*), moesin (*Msn*), myosins (*Myh10, Myl12a, Myl2, Myl9*), p21 protein (Cdc42/Rac)-activated kinase 3 (*Pak3*), platelet-derived growth factors (*Pdgfb, Pdgfc, Pdgfrb*). phosphatidylinositol 3-kinases (*Pik3ca, Pik3cb, Pik4k2a*), RAS viral (r-ras) oncogene homolog 2 (*Rras2*), vav 3 oncogene (*Vav3*), and vinculin (*Vcl*).

#### 3.3.4. Significantly Regulated Genes Responsible for the Circadian Rhythm

In the circadian rhythm pathway (Appendix A) we found as significantly up-regulated only F-box and leucine-rich repeat protein 13 (*Fbxl13*) and as down-regulated: cryptochrome 1 photolyase-like (*Cry1*), period circadian clock 2 (*Per2*), protein kinase, AMP-activated, alpha 2 catalytic subunit (*Prkaa2*). 

### 3.4. Regulation of Signaling Pathways 

Figure 4, Figure 5 and Figure 6 present the regulation of Ca^2+^-, NOD-like receptor and thyroid hormone signaling pathways. The graphs below each pathway show the expression ratios and the individual cut-offs for the pathway genes that can be considered as significantly regulated, and the GCH scores of the regulated and other important genes. Where appropriate, we have shown also the genes whose variance was so high that even though they exceeded the traditional 1.5× in gene expression ratio, the differences were not statistically significant (“false hits”).

#### 3.4.1. Significantly Regulated Ca^2+^-Signaling Genes

In Figure 4, the significantly up-regulated genes were adenosine A2 receptors (Adora2a, Adora2b), calmodulin-like 4 (Calml4), cholecystokinin B receptor (*Cckbr*), CD38 antigen (*Cd38*), endothelin receptors a/b (*Ednr/b*), erb-b2 receptor tyrosine kinase 4 (*Erbb4*), 5-hydroxytryptamine (serotonin) receptor 5B (*Htr5b*), purinergic receptor P2X, ligand-gated ion channel, 6 (*P2rx6*), phosphodiesterase 1B, Ca2+-calmodulin-dependent (*Pde1b*), phosphorylase kinases (*Phka2*, *Phkg1*), and sphingosine kinase 2 (*Sphk2*). 

Down-regulated genes were adenylate cyclase 2 (*Adcy2*), adrenergic receptor, beta 3 (*Adrb3*), arginine vasopressin receptor 1A (*Avpr1a*), calcium channel, voltage-dependent, P/Q type, alpha 1A subunit (*Cacna1a*), calcium/calmodulin-dependent protein kinase II, delta (*Camk2d*), coagulation factor II (thrombin) receptor (*F2r*), guanine nucleotide-binding proteins (*Gna14, Gnaq*), 5-hydroxytryptamine (serotonin) receptor 7 (*Htr7*), inositol 1,4,5-trisphosphate receptor 1 (*Itpr1*), purinergic receptor P2Y, G-protein coupled receptors (*P2ry1, P2ry2*), platelet-derived growth factor receptor, beta polypeptide (*Pdgfrb*), and prostaglandin F receptor (*Ptgfr*).

#### 3.4.2. Significantly Regulated NOD-Like Receptor Signaling Genes

In Figure 5, the significantly up-regulated genes were cathelicidin antimicrobial peptide (Camp), chemokine (C-C motif) ligand 5 (*Ccl5*), conserved helix-loop-helix ubiquitous kinase (*Chuk*), cathepsin B (*Ctsb*), cytochrome b-245, beta polypeptide (*Cybb*), inhibitor of kappaB kinase epsilon (Ikbke), Janus kinase 1 (*Jak1*), mitogen-activated protein kinases (*Mapk13, Mapk3*), NLR family, apoptosis inhibitory protein 2 (*Naip2*), nucleotide-binding oligomerization domain containing 1 (Nod1), pannexin 1 (*Panx1*), RanBP-type and C3HC4-type zinc finger containing 1 (*Rbck1*), TGF-beta activated kinase 1/MAP3K7-binding protein 1 (*Tab1*), toll-like receptor adaptor molecule 1 (Ticam1), and thioredoxin interacting protein (*Txnip*). 

Significantly down-regulated genes were anthrax toxin receptor 2 (*Antxr2*), B cell leukemia/lymphoma 2 (*Bcl2*), BCL2-like 1 (*Bcl2l1*), caspase 1 (*Casp*1), chemokine (C-C motif) ligand 12 (*Ccl12*), Fas (TNFRSF6)-associated via death domain (*Fadd*), guanylate-binding protein 5 (*Gbp5*), inositol 1,4,5-trisphosphate receptor 1 (*Itpr1*), mitogen-activated protein kinases (*Mapk12, Mapk8*), receptor-interacting serine-threonine kinase 3 (*Ripk3*), signal transducer and activator of transcription 2 (*Stat2*), TANK-binding kinase 1 (*Tbk1*), transient receptor potential cation channel, subfamily V, and member 2 (*Trpv2*).

#### 3.4.3. Significantly Regulated Thyroid Hormone Signaling Genes

In Figure 6, the significantly up-regulated genes were ATPase, Na^+^/K^+^ transporters (*Atp1a2, Atp1b2*), deiodinase, iodothyronine, type II (*Dio2*), forkhead box O1 (*Foxo1*), Mapk3, mediator complex subunits (*Med13l, Med24*), myosin, heavy polypeptide 6, cardiac muscle, alpha (*Myh6*), solute carrier family 2 (facilitated glucose transporter), member 1 (*Slc2a1*), and tuberous sclerosis 2 (*Tsc2*). 

Significantly down-regulated genes were ATPase, Na^+^/K^+^ transporting, alpha 4 polypeptide (*Atp1a4*), bone morphogenetic protein 4 (*Bmp4*), cyclin D1 (*Ccnd1*), catenin (cadherin-associated protein) beta 1 (*Ctnnb1*), hypoxia-inducible factor 1 alpha subunit (*Hif1a*), integrin alpha V (*Itgav*), transformed mouse 3T3 cell double minute 2 (*Mdm2*), mediator complex subunit 30 (*Med30*), phosphatidylinositol 3-kinases (*Pik3ca, Pik3cb*), regulators of calcineurin (*Rcan1, Rcan2*), solute carrier family 16 (monocarboxylic acid transporters) member 10 (*Slc16a10*), and thyroid hormone receptor beta (*Thrb*).

### 3.5. Oli-neu Proximity Remodels the Integration of Astrocytes with Neighboring, Synaptically Coupled Neurons 

As illustrated in Figure 7, KEGG pathway analysis reports that *Oli-neu* cells have an indirect modulatory role on the astrocyte response to the release of neurotransmitters. Interestingly, the astrocytes cultured in the presence of non-touching *Oli-neu* cells exhibited up-regulated genes primarily related to glutamatergic (*Glul, Grm3*), GABAergic (*Glul*), and dopaminergic (*Maob*) transmission, whereas expression of genes regarded as serotonin markers tended to be lower (*Itpr1l, Gnaq, Trpc1*). We found no significant change in the control of genes related to cholinergic synapses.

Although these astrocyte genes are mapped in the corresponding KEGG pathways, their primary functional roles are likely not in regulating synaptic activity. Thus, while *Grm3* is a metabrotropic glutamate receptor that may be involved in astrocyte-neuron signaling, *Glul* is an astrocyte biomarker and *Maob* is a marker of astrogliosis. *Itpr1* is the IP3R that is regarded as primarily neuronal. however, we found that its expression level is 2.5× higher than that of the median gene in CTR astrocytes and 1.9× in co-cultured astrocytes (significantly down-regulated by −1.3×). 

### 3.6. Cellular Environment Remodels Gene Networks

We found that, in addition to regulating numerous individual genes, proximity of *Oli-neu* cells had a major impact on the gene networks. Figure 8 illustrates this finding by the changes in the coordinated expression of *Gja1* and *Panx1* (encoding the major gap junction channel-forming protein Cx43 and the ATP release channel protein Panx1) with actin-cytoskeleton (Figure 8a) and circadian rhythm (Figure 8b) genes. Both Panx1 and Cx43 are well-documented for their important roles in brain physiopathology [56,57] through networking of astrocytes and oligodendrocytes- by way of paracrine and intercellular-channel mediated pathways, respectively. This analysis shows that the expression synergism of *Panx1* with *Fgf18, Itga2b* and *Pdgfd* in isolated (control) astrocytes was reversed to an antagonistic coordination by the proximity of *Oli-neu* cells (in insert), while the antagonism of *Gja1* with *Pip4k2b* was turned into synergism. Interestingly, expression level of *Gja1* was practically not affected by the presence of *Oli-neu* cells (the 7% observed reduction is below the corresponding cut-off) but that of *Panx1* was increased by more than 42%. More interesting is that the synergistic expression of *Gja1* with the actin-cytoskeleton genes increased by 30%, the independent expression stayed the same (3.4%) and the antagonistic expression decreased by 11%. However, the proximity of *Oli-neu* cells turned the very different coordination patterns of *Gja1* and *Panx1* with circadian rhythm genes in CTR astrocytes into practically identical ones (i.e., the two genes encoding channel-forming proteins are similarly related to the circadian rhythm genes). We found that the negative coordination of *Gja1* with *Pip4k2b* (phosphatidylinositol- 5-phosphate 4-kinase, type II, beta) in CTR astrocytes was switched to a positive one in INS astrocytes. By contrast, the expression synergisms of *Panx1* with *Fgf18* (fibroblast growth factor 18), *Itga2b* (integrin alpha 2b), and *Pdgfd* (platelet-derived growth factor, B polypeptide) in CTR were reversed into antagonistic ones in INS. 

The findings that co-culture of astrocytes with *Oli-neu* cells results in transcriptomic remodeling of a number of functional pathways led us to examine whether the presence of astrocytes in co-culture with these cells would also result in remodeling of their gene networks. To test this hypothesis, we reanalyzed the expression data from a previous experiment profiling the oli-neu cultured alone or co-cultured with astrocytes in the same experimental set up [32,33]. Figure 8c,d illustrate the remodeling of the expression coordination of *Panx1* with the same actin cytoskeleton and circadian rhythm genes in the *Oli-neu* cultured alone (Oli – Ast) or with non-touching astrocytes in the neighborhood (Oli + Ast). *Gja1* coordination partners are shown for astrocytes but not for *Oli-neu* cells which do not express Cx43. 

Among the most striking differences in the co-cultured cells were that the negative coordination of *Panx1* with *Actn1* (actinin alpha 1)*, Arghef1/4* (Rho guanine nucleotide exchange factor (GEF) 1/4), *Itgb3* (integrin beta 3)*, Pxn* (paxillin) and *Vcl* (vinculin) in astrocytes that were positive in *Oli-neu* This finding indicates that, at least for the actin cytoskeleton, *Panx1* plays opposite roles in the two types of glial cells.

## 4. Discussion 

In two previously published papers, we have shown that astrocyte-conditioned medium is a major regulator of gene expression in *Oli-neu* cells, even in the absence of cytosol-to-cytosol communication via gap junction channels connecting these two cell types [32,33]. In the present study, we analyzed whether *Oli-neu* cells-conditioned medium changes significantly the astrocyte transcriptome. Together with our studies on brains of Cx43KO, Cx32KO and Cx36 mice [9,13], these data on astrocytes and *Oli-neu* cells cultured alone and co-cultured with each other show the transcriptomic integration of the brain glia. 

Our experimental results show that glial integration persists even in the absence of direct astrocyte-oligodendrocyte communication via gap junction channels. The regulation by *Oli-neu* cells of astrocyte genes that modulate neurotransmission suggests that neurons may also participate in transcriptomic integration. Such integration of neurons into the glial network is likely carried out through astrocyte-released molecular factors binding membrane receptors on neurons (e.g., [58,59,60]), interaction could also be achieved via transfer of exosomes [61].

Microarray data were analyzed from several, complementary perspectives, considering all three independent expression features that can be determined from studies incorporating four biological replicas. The study was empowered by advanced analytical approaches. For instance, as listed in Appendix A, our method eliminated the “false hits” (absolute fold-change over 1.5× but below CUT). However, we have determined that some of the “false negatives” were actually significantly regulated (absolute fold-change less than 1.5× but over CUT), listed in Appendix A. Moreover, we complemented the traditional presentation of percentage of significantly regulated genes within affected pathways with a metric (Weighted Pathway Regulation; WPR) that weighs the contribution of each regulated gene with respect to a cut-off tailored for it by considering the technical noise and biological variability. Of note in Figure 1a is that, from the perspective of the WPR score, the gap junction (GJ) pathway was more affected than the others (WPR = 43.3), indicating that *Oli-neu* proximity exerted a major impact on genes involved in hetero-cellular communication even in the absence of cell-to-cell contact. 

In the case of the cell cycle, the much larger percentage of down-regulated genes suggests a slow-down of the mitotic progression of astrocytes in the vicinity of oligodendrocytes that we have observed but have not yet quantified. Out of five quantified cyclin-dependent kinases that control cell progression through cell cycle owing to their activation of cyclins by catalytic CDK, three (*Cdk1, Cdk2, Cdk6*) were significantly down-regulated while the other two (*Cdk4, Cdk7*) were not affected in co-culture. Moreover, only one (*Cdkn2a*) of six cyclin-dependent kinase inhibitors was down-regulated and the others not changed. Also, the transcription factors *E2f1*, *E2f2* and four (*Mcm2, Mcm4, Mcm5, Mcm6*) mini-chromosome maintenance deficient homologs were down-regulated. Interestingly, two isoforms of the MAD2 mitotic arrest deficient-likes (*Mad2l1* −1.48×; *Mad2l2* 1.44×) and two isoforms of the beta transforming growth factors (*Tgfb1* 1.32×, *Tghb3* −0.39×) are symmetrically regulated, likely balancing their effects on the cell cycle. 

Relative Expression Variability (REV) quantifies the vulnerability of the gene expression in response to even subtle environmental changes. Very low REV (expression level fluctuates within very narrow interval across biological replicates) indicates strong control of transcript abundance exerted by homeostatic mechanisms, most likely reflecting survival advantage afforded by proportional expression levels of the genes. Whereas strictly controlled genes likely reflect those that are critical for cell survival or phenotypic expression, genes with high REV may represent ones that are more susceptible to adaptation to environmental fluctuations. 

In several other transcriptomic studies, we found an overall reduction of gene REV in cells and tissues collected from subjects with various diseases (epilepsy [62], experimental autoimmune encephalomyelitis [63], kidney cancer [64]) compared to healthy counterparts. REV was also significantly lower in tissues from animals subjected to various stresses (microgravity [65], chronic hypoxia [66,67,68]) or genetic manipulations (e.g., [8,37]. In this context, the overall higher expression variability seen in astrocytes with *Oli-neu* cells in the neighborhood suggests that co-culture more closely approximates conditions in the brain.

We calculated a parameter (GCH) to assess the degree to which expression of a gene influences the expression stability and interconnectivity of the functional pathway in a specified condition. While the GCH analysis confirmed the known impact of some genes, for others additional experiments may be needed to understand their role in the pathway. Since GCH was derived from measures of expression control and coordination that are independent between each other and from the average expression levels, correspondence between genes with high GCH in one condition is not expected to predict significant regulation in another. However, experimental alteration of a gene is expected to have transcriptomic consequences in line with its GCH. Interestingly, the top gene (highest GCH) in one condition scores very low in the other. Thus, *Serp2* has GCH = 49.31 in CTR astrocytes but GCH = 2.49 in INS astrocytes, while *Pdcd7* has GCH = 48.13 and GCH = 2.14 in CTR astrocytes. Owing to its critical role for the regulation of transcription and translation to protect against ER stress caused by the accumulation of unfolded proteins [69,70], the high level of *Serp2* is consistent with higher stress levels in astrocytes cultured alone. The substantial differences in GCH values for *Serp2* between the two conditions is in agreement with the conclusion above that the overall lower REV for astrocytes cultured alone may reflect higher stress levels. The programmed cell death 7 gene (*Pdcd7)* is much more highly ranked in co-cultures than in astrocytes alone (GCH = 48.13 in INS astrocytes compared to GCH = 2.14 in CTR astrocytes). This may reflect inhibition of astrocyte proliferation [71] that occurs during brain development and is consistent with the s their observed reduced proliferation and down-regulation of the cell-cycle pathway in co-culture. 

The prominence of *Tubb3* in the GJ pathway of CTR astrocytes confirms its role in cytoskeletal organization of microtubules in astrocytes cultured without any other cells [72], while that of *Sos*1 in co-cultured astrocytes shows its prevalence with regard to cell proliferation and viability [73]. Interestingly, all three cell-cycle genes in the *Ccnh-Cdk7=Pea15a* complex [74] of the basal transcription factor TFIIH have almost equally low positions in astrocytes cultured alone (GCH_CTR_ = 1.36 (*Ccnh*), 1.63 (*Cdk7*), 1.88 (*Pea15a*)). However, in the presence of *Oli-neu* cells *Ccnh* is promoted to the top position in this pathway (GCH_INS_ = 12.01), while the rankings of the other genes of the complex were largely unchanged: 1.45 (*Cdk7*), 1.65 (*Pea15a*)). The high position of *Ccnh* in an environment closer to that of the brain can explain why the Ex8+49T>C variant of this gene is associated with an increased risk of glioma [75]. 

We found that several members of the gap junction pathway-related genes (including *Gjb2*, encoding Cx26) were differentially expressed in astrocytes in the presence of *Oli-neu* cells (Figure 2). However, the expression of the main connexin gene, *Gja1*, was not affected. This result is surprising because the oligodendrocytes responded to the presence of astrocytes by increasing the expression of *Gjc2* (encoding Cx47, the Cx43 partner in heterocellular gap junction channels with oligodendrocytes identified in the brain [76], by 9.7×. In our study of the impact of astrocyte proximity on *Oli=neu* cells, we reported [32] an increased number of synergistically expressed partners of *Gjc2* with myelination genes by almost ten-fold (from 2.5% to 22.4%). This result suggests that adjacent astrocytes may prepare immature oligodendrocytes for their main function, to myelinate the axons. 

We were surprised to find through KEGG analysis that the proximity of oligodendrocytes regulated genes involved in interactions of astrocytes with neuronal synapses [77,78,79]. Although the association of these genes with those pathways may or may not indicate their primary functional roles, the finding does show that co-culture of astrocytes with cells that have some characteristics of oligodendrocytes potentially impacts neuronal components of transcriptomic networks in glia. 

While the cells in these experiments are not able to interact via gap junctions, it becomes all the more interesting that the gene networking interactions by the Cx43 and Panx1 genes reconfigure in response to the changed cellular environment represented by non-contacting *Oli-neu* cells. The coordination patterns of *Panx1* illustrate the degree to which gene networks can be remodeled in response to environmental factors. *Panx1* has mostly synergistic coordination with actin cytoskeleton genes in astrocytes cultured alone, whereas it is antagonistically coordinated with these genes when cocultured with *Oli-neu cells* (Figure 8a). Conversely, *Oli-neu cells* actin genes are largely independent of *Panx1* when cultured alone but become synergistically coordinated with *Panx1* when cultured with astrocytes (Figure 8d). The opposite consequences of coculture on *Panx1*-coordinated cytoskeleton-related genes in astrocytes and *Oli-neu cells* might reflect fundamental divergence in maturation or/and response to inflammatory or other stimuli. 

A distinct type of network remodeling is exemplified by the mostly antagonistic coordinations of *Gja1* with circadian rhythm genes in cultured alone astrocytes, with almost no coordination of *Panx1* with genes in this pathway. In coculture, antagonistic and synergistic coordinations of CR genes with *Gja1* are approximately equal, while each coordination with *Gja1* is matched by a similar coordination with *Panx1*. Thus, these two large conduction channel proteins act independently on circadian rhythm genes of astrocytes alone but become cooperative and may compensate for one another when in the presence of *Oli-neu cells.* Interestingly, we reported that the coordination patterns of *Gja1* and *Panx1* with the whole brain transcriptome of wild type mice were extraordinarily (90.8%) similar [11]. 

The general trend with regard to circadian genes is down-regulation of those that control circadian cycling of downstream transcriptional regulators and up-regulation of the circadian cycle dampening gene *Fbxl13* (F-box and leucine-rich repeat protein 13). *Fbxl13* has been implicated in increasing ubiquitination and degradation of cryptochrome proteins, including *Cry1*, through ubiquitin-protein ligase activity of Fbxl13 [80]. We found that the presence of nearby *Oli-neu cells* significantly up-regulated *Fbxl13* by 2.6×, down-regulated *Cry1* by −1.2×, and left *Cry2* unchanged. Effects on the circadian rhythm pathway predict that rhythms might be either more rapid or dampened. This expectation is born out in studies in mice in which knockout of Cx43 (along with Cx30) in astrocytes disrupted circadian rhythm and produced higher circadian rhythm amplitude [81]. The finding of increased expression of *Gjb2* and *Fbxl13* in astrocytes grown in the presence of *Oli-neu* cells might be a starting point for understanding the role of astrocyte connexins in controlling circadian rhythm. 

The perivascular end-feet of astrocytes have been implicated in the uptake of thyroid hormones, especially thyroxine (T4) which is the vastly predominant, but mostly inactive form of thyroid hormone in the blood. After T4 is transported through brain endothelial cells it is thought to be mainly brought into astrocytes by transporters with varying specificity for T4. Once taken into astrocytes, T4 is converted to the highly active form triiodothyronine (T3) by iodothyronine deiodinase type 2 (DIO2) with *DIO2* mRNA expression largely exclusive to astrocytes as reviewed in [82]. Taken together, the preceding aspects of thyroid handling point to astrocytes as key uptake and distribution cells in the brain. Additionally, oligodendrocyte maturation and key myelin production gene expression is strongly regulated (promoted) by T3 [83,84] Therefore, thyroid hormone handling pathway is important to oligodendrocytes and their precursors leading us to examine our data for changes astrocyte thyroid pathway genes in response to the presence of non-contacting *Oli-neu* cells. The presence of *Oli-neu* cells decreased expression of *Slc16a10* (*Mct10*) but did not produce a change in the other two transporters expressed in astrocytes: *Slo1c1* (OATP1C1) and *Slc16a2* (MCT8). More interestingly, *Dio2* was up-regulated in the presence of *Oli-neu* cells. *Tbc1d4* is up-regulated in astrocytes cultured in the presence of *Oli-neu* cells and this GTPase increases surface expression of glucose transporters [85]. These results may indicate a shift towards an astrocyte phenotype that may produce activated thyroid hormone and bring in additional glucose. It is interesting to speculate that the presence of *Oli-neu* cells produces gene expression changes in astrocytes that would support myelination through increased production of activated T3 and support of the high metabolic demands of myelinating oligodendrocytes. 

## 5. Conclusions

The major caveat of this study is that the *oli-neu* cells are not primarily differentiated oligodendrocytes, but rather were derived from immortalization of oligodendrocyte precursor. The use of a cell line for the cocultures standardized the donor cell population, thus limiting the variability so that comparisons with high significance could be obtained from astrocytes derived from mouse brains using our analytical methods. Our results clearly indicate that the cellular environment remodels astrocyte transcriptomic networks by modulating the expression level, the expression control, and the expression coordination of genes in each of the pathways that we analyzed. Our previous analogous study of *Oli-neu* gene expression alterations caused by the co-culture of astrocytes [32,33] also revealed substantial overall changes. We thus predict that astrocytes and other cell types within the complex heterocellular brain will also be quite sensitive to environmental factors released by neighboring cells, which include not only soluble growth factors and other hormones but also lipids, nucleotides, and exosomes. Although our study exemplifies how transcriptomic network dynamics may be mechanistically dissected, the intricate multilevel interdependencies complicate the analysis of such changes at the level of a heterocellular tissue such as the brain.

## Figures and Tables

**Figure 1 genes-11-00520-f001:**
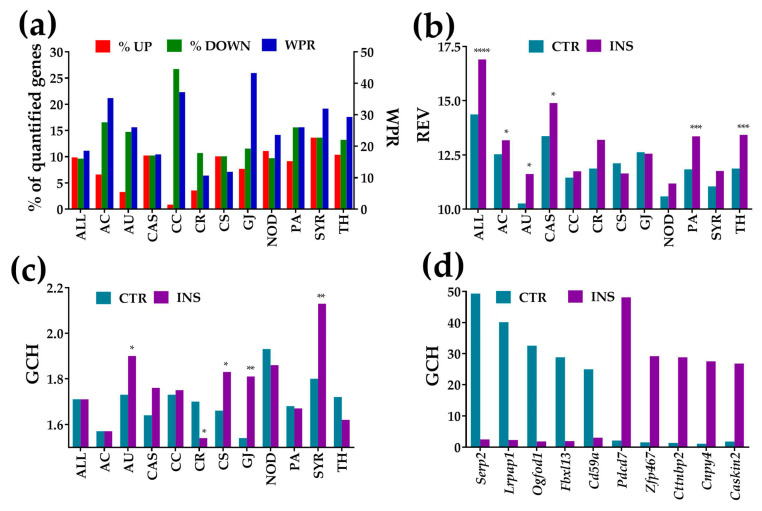
Alterations of the expression level, expression variability, and gene commanding height for all 18,891 quantified unigenes (ALL) and selected functional pathways and individual genes. (**a**) Percentages of significantly up/down-regulated genes and the Weighted Pathway Regulation (WPR) score of the selected pathways in astrocyte co-cultured with *Oli neu* cells (INS) with respect to astrocytes cultured alone (CTR). (**b**) Median Relative Expression Variability (REV) of selected gene groups in each condition. (**c**) Median Gene Commanding Height (GCH) of selected gene groups in each condition. (**d**) Top 5 genes (highest GCH scores) in each condition and their GCHs in the other condition. Statistical significance: * *p* < 0.05, ** *p* < 0.01, *** *p* < 0.005, **** *p* < 0.0001.

**Figure 2 genes-11-00520-f002:**
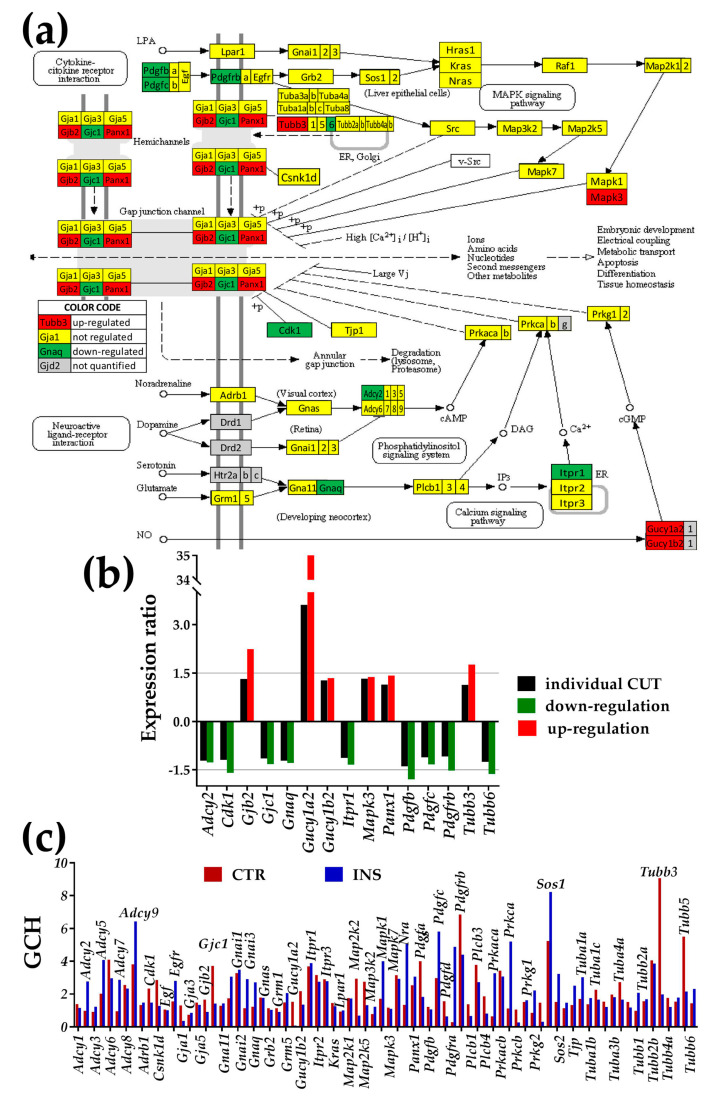
Presence of non-touching *Oli-neu* cells affects genes in the gap junction (GJ) pathway (modified from https://www.genome.jp/kegg-bin/show_pathway?mmu04540). (**a**) Regulation (up-red background, down-green background) of the interconnected genes within the GJ pathway. (**b**) Expression ratios and individual fold-change cut-offs (both negative for down-regulation) of the significantly regulated genes. (**c**) Gene Commanding Height (GCH) scores of the GJ genes in the two conditions. The most prominent gene in CTR astrocytes is *Tubb3* (GCH = 9.07), while in INS cells it is *Sos1* (GCH = 8.23). As illustrated in Figure 1c, considering the entire pathway, the median GCH increased by over 17% in INS with respect to CTR.

**Figure 3 genes-11-00520-f003:**
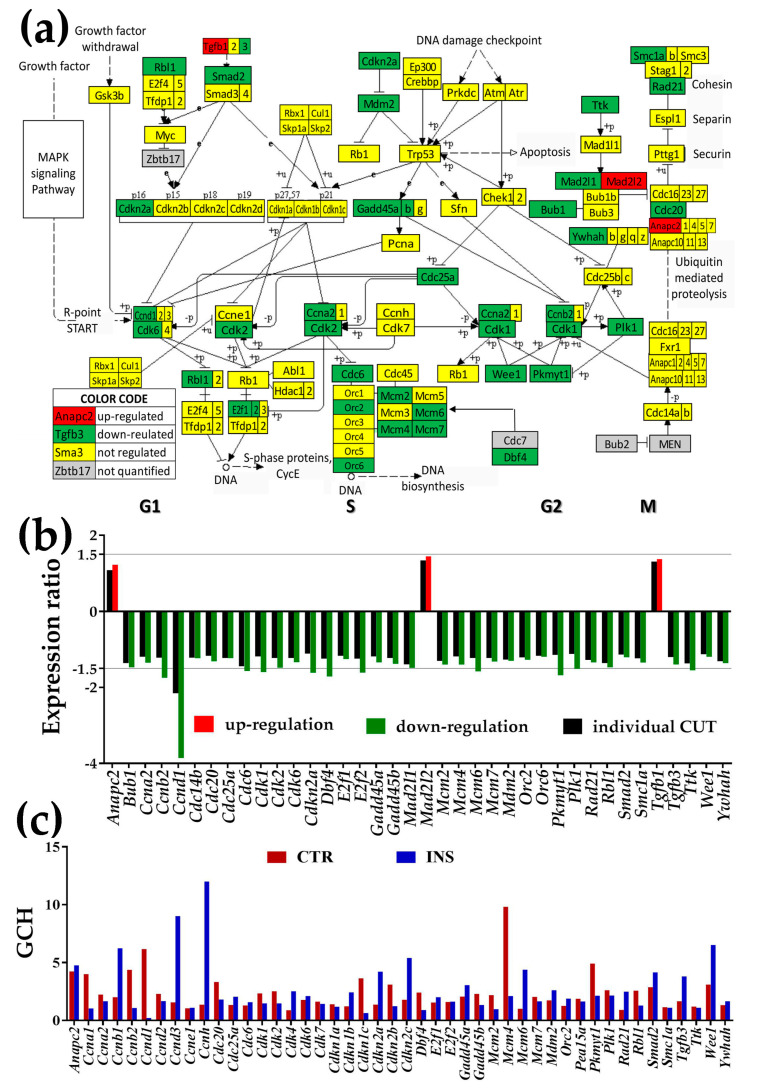
Presence of non-touching *Oli-neu* cells affects cell-cycle (CC) pathway genes (modified from https://www.genome.jp/kegg-bin/show_pathway?mmu04110). (**a**) Regulation of the interconnected genes within the CC pathway. (**b**) Expression ratios and individual fold-change cut-offs (negative for down-regulation) of the significantly regulated genes. Note that the individual gene cut-off for *Ccnd1* (CUT = 2.152) exceeded 1.5×. (**c**) Gene Commanding Height (GCH) scores of the significantly regulated CC genes and other quantified cyclins (*Ccna1, Ccnb1, Ccnd2, Ccnd3, Ccne1, Ccnh*) and kinase inhibitors (*Cdkn1a, Cdkn1b, Cdkn1c, Cdkn2b, Cdkn2c). Mcm4* (GCH = 9.81) was the highest ranked CC gene in CTR and Ccnh (GCH = 12.01) the highest ranked CC gene in cocultured with *Oli-neu* astrocytes.

**Figure 4 genes-11-00520-f004:**
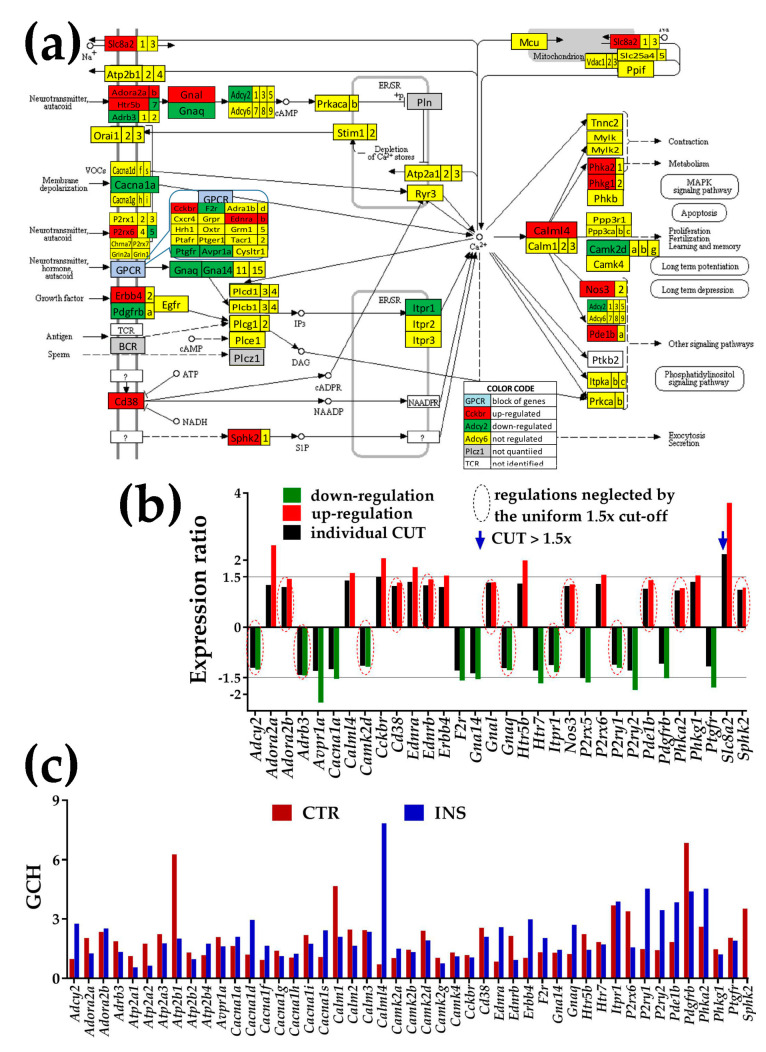
Presence of non-touching *Oli-neu* cells regulates calcium signaling (CAS) pathway in astrocytes (modified from mmu04020, www.kegg.jp). (**a**) Regulation of interconnected genes within the CAS pathway. (**b**) Expression ratios and individual fold-change cut-offs (negative for down-regulation) of the significantly regulated genes. Note that 14 out of 32 significantly regulated genes had absolute fold-changes below the traditional 1.5× and that CUT exceeded 1.5× for *Slc8a2.* The regulated below 1.5× include: adenylate cyclase 2 (*Adcy2*), adenosine A2b receptor (*Adora2b*), adrenergic receptor beta 3 (*Adrb3*), calcium/calmodulin-dependent protein kinase II, delta (*Camk2d*), CD38 antigen (*Cd38*), endothelin receptor type B (*Ednrb*), guanine nucleotide-binding proteins (*Gnal*, *Gnas*), inositol 1,4,5-trisphosphate receptor 1 (*Itpr1*), nitric oxide synthase 3, endothelial cell (*Nos3*), purinergic receptor P2Y, G-protein coupled 1 (*P2ry1*), phosphodiesterase 1B, Ca^2+^-calmodulin-dependent (*Pde1b*), phosphorylase kinase alpha 2 (*Phka2*), and sphingosine kinase 2 (*Sphk2*). (**c**) Gene Commanding Height (GCH) scores of selected CAS genes. Owing to their recognized importance, in (**c**) the set of the significantly regulated CAS genes was completed with: Ca^2+^-transporting ATPases (*Atp2a1, Atp2a2, Atp2a3, Atp2b1, Atp2b2, Atp2b4*), calcium channels (*Cacna1d, Cacna1f, Cacna1g, Cacna1h, Cacna1i, Cacna1s*), calmodulins (*Calm1, Calm2, Calm3*), and calcium/calmodulin-dependent protein kinases (*Camk2a, Camk2b, Camk2g, Camk4*).

**Figure 5 genes-11-00520-f005:**
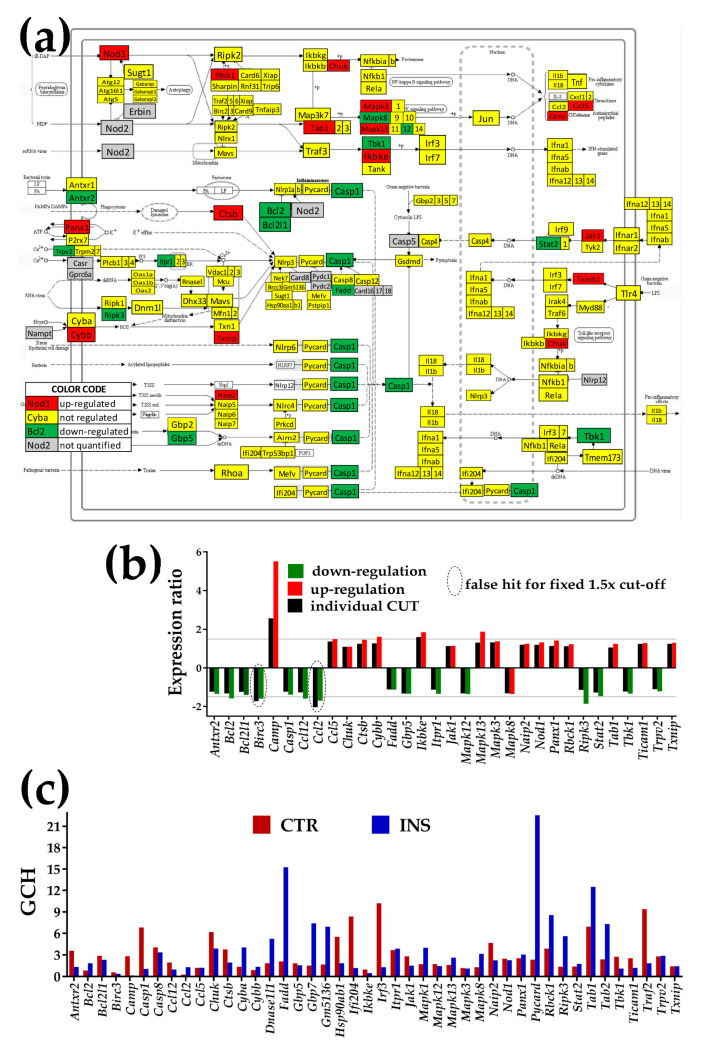
Proximity of non-touching *Oli-neu* cells affects the NOD-like receptor signaling pathway (modified from http://www.kegg.jp/pathway/mmu04621). (**a**) Regulation of interconnected genes. (**b**) Expression ratios and individual fold-change cut-offs (negative for down-regulation) of the significantly regulated genes. Note that *Birc3* (baculoviral IAP repeat-containing 3) and *Ccl12* (chemokine (C-C motif) ligand 2) would be false hits if the traditional 1.5× were used instead of our individual gene cut-off because although their expression ratios were over the 1.5× limit, they did not exceed their individual CUT. (**c**) Gene Commanding Height (GCH) scores of selected NOD genes. The set of the significantly regulated NOD genes was completed with genes having GCH over 4 in either condition: *Casp8, Cyba, Dnase1l1, Gbp7, Gm5136,* heat shock protein (*Hsp90aa1*), interferon regulatory factor (*Irf3*), mitogen-activated protein kinase (*Mapk1*), PYD and CARD domain containing (*Pycard*, the most prominent of the NOD pathway in the INS astrocytes), *Tab2, Traf2*.

**Figure 6 genes-11-00520-f006:**
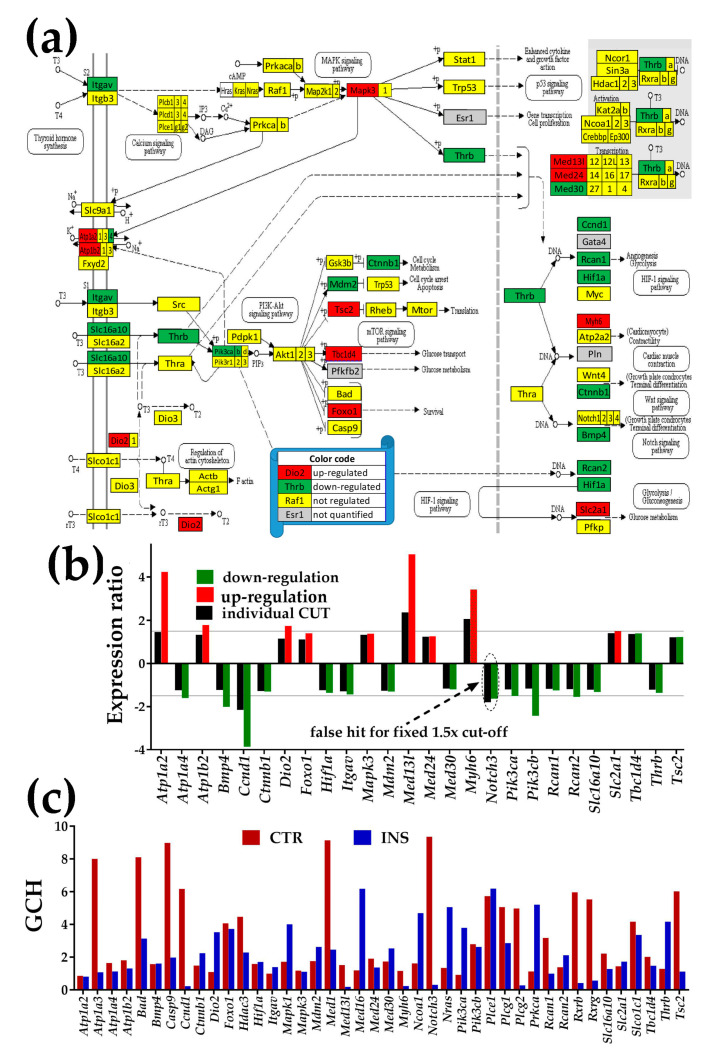
Proximity of non-touching *Oli-neu* cells affects the thyroid hormone signaling pathway (modified from http://www.kegg.jp/pathway/mmu04919). (**a**) Significantly regulated genes. (**b**) Expression ratios and individual fold-change cut-offs (negative for down-regulation) of the significantly regulated genes. Note that *Notch3* would be a false hit if the traditional 1.5× would be used instead of our individual gene cut-off because although the absolute expression ratio exceeded 1.5× it was below the individual gene CUT. (**c**) Gene Commanding Height (GCH) scores of selected TH genes. In addition to the significantly regulated TH genes, we plotted genes with high GCH scores in one condition: Na^+^/K^+^ transporting, alpha 3 polypeptide (*Atp1a3*), B cell leukemia/lymphoma 2 (BCL2)-associated agonist of cell death (*Bad*), caspase 9 (Casp9), subunits of mediator complex (*Med1, Med16*), notch 3 (*Notch3*), parathyroid hormone 1 receptor (*Pth1r*), regulators of calcineurin (*Rcan1, Rcan2*), and thyroid hormone receptor-associated protein 3 (*Thrap3*).

**Figure 7 genes-11-00520-f007:**
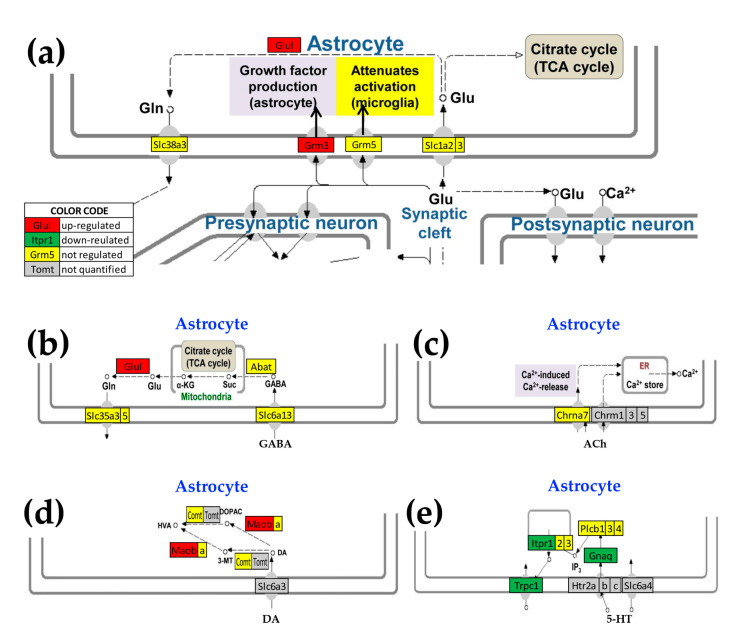
Astrocyte genes designated by KEGG pathways as corresponding to synaptic neurotransmission that were altered by the presence of *Oli-neu cells* (**a**) Astrocyte genes encoding proteins that interact with those in the pre- and post-synaptic. neuronal glutamatergic synapses (modified from mmu04724, kegg.jp), (**b**) astrocyte genes related to GABAergic synapses (modified from mmu04727, kegg.jp), (**c**) cholinergic synapse (modified from mmu04725, kegg.jp), (**d**) dopaminergic synapse (modified from mmu04728, kegg.jp), (**e**) serotonergic synapse (modified from mmu04726, kegg.jp). Genes: 4-aminobutyrate aminotransferase (*Abat*), cholinergic receptor, nicotinic, alpha polypeptide 7 (*Chrna7*), catechol-O-methyltransferase (*Comt*), glutamate-ammonia ligase (*Glul*), guanine nucleotide-binding protein, alpha q polypeptide (*Gnaq*), metabotropic glutamate receptors (*Grm3, Grm5*), inositol 1,4,5-trisphosphate receptors (*Itpr1, Itpr2, Itpr3*), monoamine oxidases (*Maoa, Maob*), beta phospholipases C (*Plcb1, Plcb2, Plcb3*), solute carriers (*Slc1a2, Slc1a3, Slc35a3, Slc35a5, Slc38a3, Slc6a13*), and transient receptor potential cation channel, subfamily C, member 1 (*Trpc1*).

**Figure 8 genes-11-00520-f008:**
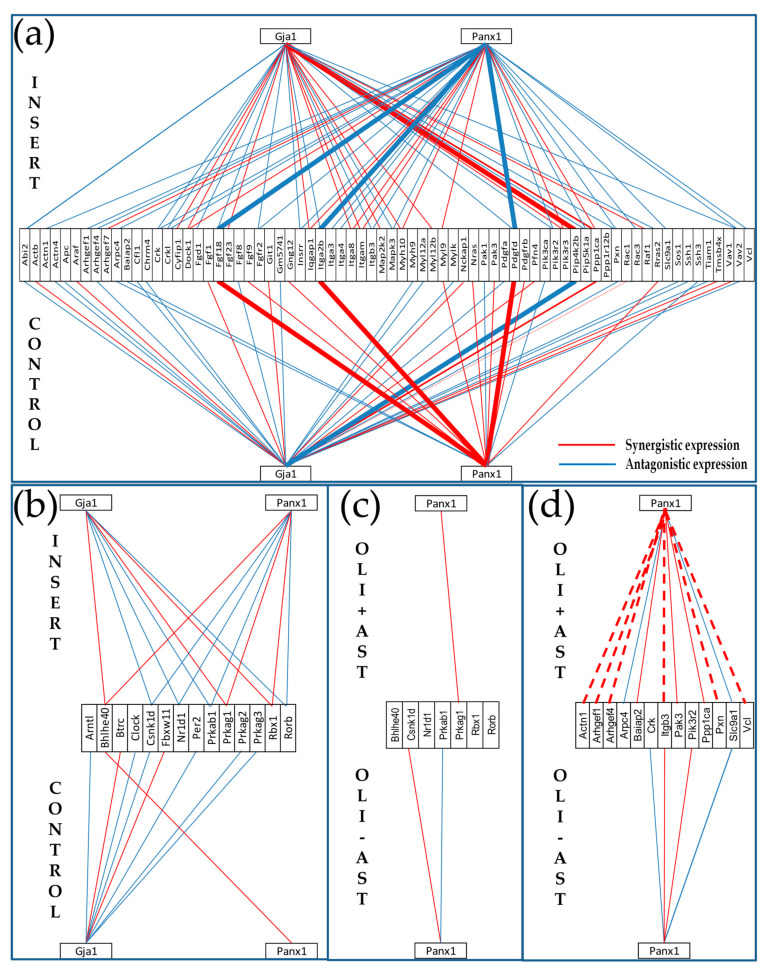
Topology of gene networks is sensitive to the cellular environment. (**a**,**b**) Proximity of non-touching olineu remodels the transcriptomic networks by which expression of genes encoding connexin43 (*Gja1*) and pannexin1 (*Panx1*) relate to the expression of (**a**) actin cytoskeleton and (**b**) circadian rhythm genes in astrocytes. (**c**,**d**) Proximity of non-touching astrocytes changes the expression coordination of *Panx1* with (**c**) circadian rhythm and (**d**) actin cytoskeleton genes in *Oli-neu*. Thicker continuous lines in (**a**) indicate reversal type of coordination in astrocyte by oligodendrocyte proximity. Dashed lines in (**d**) indicate opposite coordination in astrocytes and oli neu when co-cultured. **CONTROL/OLI-AST** = astrocytes/*Oli-neu cells* cultured alone. **INSERT/OLI+AST** = co-culture of astrocytes and *Oli-neu cells*. Note: in order to simplify the illustration, only the *Panx1* coordination partners from the two pathways in astrocytes are presented in the *Oli-neu* cells.

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
