# Peer review of "Cellular Environment Remodels the Genomic Fabrics of Functional Pathways in Astrocytes"

_genes, 2020, doi:10.3390/genes11050520_

Round 1

Reviewer 1 Report

The original paper entitled « Oligodendrocytes remodel the genomic fabrics of functional  pathways in astrocytes » provides in vitro evidence that oligodendrocytes (precursor) remotely influence astrocyte biology and transcriptome through putative soluble factors.

General comments:

The manuscript is well illustrated with underlined transcriptomes and analysis of functional astrocyte pathways; although some parts of the results, including lists of genes differentially regulated, remain very descriptive, and sometimes exhaust the attention span of the reader. The results report array findings about differences in gene expressions up- or down-regulated in astrocytes when in presence or absence of uncoupled oligodendrocytes.

Major remarks:

- The method for assessing the purity of astrocyte cell culture is not mentioned. Given the focus on specific astrocyte transcriptome and the modest number of replicates per experimental group, I think that this type of information is highly valuable.

- The term “oligodendrocytes” used in the title and the abstract is a bit misleading because throughout the study, as stated in the discussion, were used a co-culture with an immortalised cell line of OL precursors. The fact that the cells are immortalized does not bother me that much, I’m more concerned by the state of differentiation of those cells compared to mature myelinating OLs.

How far or how close are those olig-neu cells from primary isolated OLs or from OPC-derived OLs, in terms of transcriptional profiling?

- Do authors have gene expression data comparing the setting with the astrocytes stimulated by OL precursors in the insert chamber AND astrocytes stimulated with the OLs conditioned medium? It would be interesting to present those data and emphasize how the A/O crosstalk influences mutually the transcriptome and behaviour of both cell types, in absence of any intercellular junctions.

- Surprisingly, the expression level of Gja1 was (significantly) not modulated by the presence of OLs. Another explanation not raised in the discussion is maybe the high basal expression of Cx43 from astrocyte-astrocyte homotypic gap junctions. Do authors have clue whether astrocytes in their lower chamber make intercellular connections and of what type? Or the astrocytes are grown such a way they are maintained under the confluency? This kind of information is also of interest because it can impact on astrocyte coupling and gene-related pathways. Connexin 43 trafficking (from cytosolic to membrane –bound pools) is another variable that should be discussed (Epifantserva et al., BBA, 2019).

- The abstract is poorly informative regarding the original data generated from gene and pathways analysis, instead there is more emphasize about the previous studies on Cx43, Cx32 KO cells …

Minor remarks:

- Even at low magnification, some cell’s aspects comparing cultivated cells would be welcome to reinforce diagrammatic data, but, probably not necessary as the specialized journal ‘Genes’ concerned mainly molecular aspects.

Typographic

page 2, line 63:    reference ‘8’ is probably [28]

page 10, line 338 : figuure

page 15, line 420 : oligodendryocytes

page 18 

line 486: adding a word to comprehend the comparison between astrocytes and oligodendrocytes as… ‘that were positive’ instead of ‘astrocytes were positive’ 

lines 498-501:    lack of ‘.’ Between sentences.

lines 500-501:

  Gap junctions between neurons and astrocytes or oligodendrocytes?   I though…t we have shown those and referenced.

Reviewer 2 Report

The study by Iacobas and colleagues investigates the impact of oligodendrocyte-like cells upon astrocyte transcriptome. They find the presence of oligodendrocytes significantly changes the transcriptome of cultured astrocytes. The conclusion is that oligodendrocytes can alter the transcriptome of neighbouring astrocytes, both in profile and also gene networks. However, considering the various limitations of cell culture (embryonic/neonatal cells or immortalised cells, artefactual culture media and growth conditions, 2D environment, lack of other cell types, etc), the specific changes identified may only be relevant for this exact model system. Furthermore, given the limitations of transcriptomics, any changes would need to be confirmed at a functional level. These caveats should be described.

A subtle issue extending throughout the manuscript is the focus on gap junctions even though the study does not model this in any manner. This is exclusively a study of the impact of oligodendrocyte secretome on astrocyte transcriptome, completely independent of gap junctions. Discussion of gap junction communcation throughout is not currently framed with this in mind. Related to this, no mention of exosomal communication. Also, as the study design utilises oligodendrocyte secretome, it reflects the astrocyte response to generalised signals from oligodendrocytes which could also influence other neighbouring cells, regardless of cell type, rather than the specific communication between oligodendrocytes and astrocytes (as would occur through gap junctions). Specific gap junction communication could have substantially different impact.

Although unclear, it appears that the entire study is conducted on a single biological replicate, with four technical replicates (separate wells) for each condition. This should be explicitly stated, and described as a caveat of the study.

In Discussion, authors conclude that higher REV of astrocytes co-cultured with oligodendrocytes indicates a closer to physiological environment. Based on the preceding discussion, a more accurate conclusion would be a less stressed environment.

The conclusion of discussions of GCH for Serp2 and Pdcd7 simply reiterates the main function of these genes, rather than the impact of altered expression in presence of oligodendrocytes. For example, rather than suggest that the high level of Serp2 plays a critical role in preventing ER stress, perhaps describe that the high level of Serp2 suggests astrocytes cultured alone have higher level of (or higher potential for) ER stress than in presence of oligodendrocytes. Similar for Pdcd7.

While interesting, the relevance of changes in expression of synaptic transmission genes due to presence of oligodendrocytes is questionable given the likely stronger influence of the presence (and activity) of neurons themselves on these genes.

The conclusion that the current study “clearly indicate that cellular environment plays [a role] in each and every cell phenotype [in the brain]” is unsubstantiated. Although possible and even likely, this claim is not supported by the current study. It can only be concluded that oligodendrocytes and astrocytes influence each other’s transcriptomes.

Fig 4b: Genes for which ‘regulations neglected by the uniform 1.5x cut-off’ are highlighted, but not mentioned in the text, and although many genes meet this criteria, they are not similarly highlighted in other figures. Also, some figure legends describe these genes while other figures do not. Suggest altering for consistency across all analyses and data presentations.

Instead of ’bullet points’ in the text, authors could consider inclusion of a table to summarise the pathways interrogated and the rationales for their inclusion.

The study amounts to transcriptomic analysis of a single co-culture of astrocytes and oligodendrocyte-like cells. Given the potential variability of cultures, it would be helpful to include images of the cells utilised.

- Gene names must be italicised in section 3.3; Fig 1d, 2bc, 3bc, 4bc, 5bc, 6bc, S1bc, S2bc; Tables S1, 2, 3; Figure 3 legend.

- Centrifuge speed should be given in g or RCF, not RPM.

- Page 8, line 284: Guc1a2 should presumably be Gucy1a2

- Page 10, line 324: Out of place text?

Reviewer 3 Report

This is a solid manuscript describing the bi-directional influence of two glial cell types on their respective gene expression profiles in vitro. I commend the authors on clear explanations on their analysis decisions.  However, I do have strong concerns about the biological interpretations that are present in the manuscript, while acknowledging that this is clearly not the authors' main focus.  Overall, I believe this manuscript describes phenomena that can exist, but does not show evidence that it does outside of the closed experimental system. This is a major limitation.

Manuscript changes needed to make these caveats clearer earlier:

  1. Use of the oli-neu cell line. These are NOT oligodendrocytes. While acknowledged clearly in the conclusion of the manuscript, I recommend that use of 'oligodendrocyte(s)' when used in the text and referring to the experiments performed be replaced with 'oli-neu' cell line (ie. mostly results and methods section). This is because it is very likely that the differentiation state of oligodendroglia will influence regulation of the corresponding astrocyte behaviour (and vice versa) and this is not addressed (or discussed). This caveat to the interpretation of the results needs to be upfront and clear from the start, not only addressed at the end.

Related to this, the stage of differentiation the oli-neu cell most resembles (presumably OPCs) should be mentioned briefly.

Minor comments

  1. The term "transcriptomic continuity" is used heavily in the introduction but is not defined and is not used after. This is jargon and it is not entirely clear what the authors truly mean. The context in which appears implies that they mean to use it to indicate that oligodendrocyte expression profiles become more astrocytic in gene expression profiles without the connexins and this doesn't not seem be related to their data. 
  2. Section 59-63; the emergent roles that oligodendroglia have in these biological processes are not noted here. Why? The presentation of oligodendroglia as almost inert cells seems misleading and counter to the experimental design.
  3. Reactive state of cultured oligodendrocytes -- while briefly signaled to in the discussion it would appear to be pertinent information to be noted in the results section. 

Round 2

Reviewer 1 Report

The authors have adequately addressed all the comments.